# Predictive Value of Serum Ferritin in Combination with Alanine Aminotransferase and Glucose Levels for Noninvasive Assessment of NAFLD: Fatty Liver in Obesity (FLiO) Study

**DOI:** 10.3390/diagnostics10110917

**Published:** 2020-11-08

**Authors:** Cristina Galarregui, Bertha Araceli Marin-Alejandre, Nuria Perez-Diaz-Del-Campo, Irene Cantero, J. Ignacio Monreal, Mariana Elorz, Alberto Benito-Boillos, José Ignacio Herrero, Josep A. Tur, J. Alfredo Martínez, M. Angeles Zulet, Itziar Abete

**Affiliations:** 1Department of Nutrition, Food Sciences and Physiology and Centre for Nutrition Research, Faculty of Pharmacy and Nutrition, University of Navarra, Irunlarrea 1, 31008 Pamplona, Spain; cgalarregui@alumni.unav.es (C.G.); bmarin.1@alumni.unav.es (B.A.M.-A.); nperezdiaz@alumni.unav.es (N.P.-D.-D.-C.); icgonzalez@unav.es (I.C.); jalfmtz@unav.es (J.A.M.); 2Navarra Institute for Health Research (IdiSNA), 31008 Pamplona, Spain; jimonreal@unav.es (J.I.M.); marelorz@unav.es (M.E.); albenitob@unav.es (A.B.-B.); iherrero@unav.es (J.I.H.); 3Clinical Chemistry Department, Clinica Universidad de Navarra, 31008 Pamplona, Spain; 4Department of Radiology, Clinica Universidad de Navarra, 31008 Pamplona, Spain; 5Liver Unit, Clinica Universidad de Navarra, 31008 Pamplona, Spain; 6Centro de Investigación Biomédica en Red de Enfermedades Hepáticas y Digestivas (CIBERehd), 28029 Madrid, Spain; 7Biomedical Research Centre Network in Physiopathology of Obesity and Nutrition (CIBERobn), Instituto de Salud Carlos III, 28029 Madrid, Spain; pep.tur@uib.es; 8Research Group on Community Nutrition and Oxidative Stress, University of Balearic Islands & Balearic Islands Institute for Health Research (IDISBA), 07122 Palma, Spain

**Keywords:** fatty liver, insulin resistance, iron, obesity, biomarker, metabolism

## Abstract

The identification of affordable noninvasive biomarkers for the diagnosis and characterization of nonalcoholic fatty liver disease (NAFLD) is a major challenge for the research community. This study aimed to explore the usefulness of ferritin as a proxy biomarker of NAFLD condition, alone or in combination with other routine biochemical parameters. Subjects with overweight/obesity and ultrasound-confirmed liver steatosis (*n* = 112) from the Fatty Liver in Obesity (FLiO) study were assessed. The hepatic evaluation considered magnetic resonance imaging, ultrasonography, and credited routine blood liver biomarkers. Anthropometry and body composition, dietary intake (by means of a validated 137-item food frequency questionnaire), and specific biochemical markers were also determined. Serum ferritin levels were analyzed using a chemiluminescent microparticle immunoassay kit. Lower serum ferritin concentrations were associated with general better liver health and nutritional status. The evaluation of ferritin as a surrogate of liver damage by means of quantile regression analyses showed a positive association with alanine aminotransferase (ALT) (β = 19.21; *p* ≤ 0.001), liver fat content (β = 8.70; *p* = 0.008), and hepatic iron (β = 3.76; *p* ≤ 0.001), after adjusting for potential confounders. In receiver operating characteristic (ROC) analyses, the panel combination of blood ferritin, glucose, and ALT showed the best prediction for liver fat mass (area under the curve (AUC) 0.82). A combination of ferritin and ALT showed the higher predictive ability for estimating liver iron content (AUC 0.73). This investigation demonstrated the association of serum ferritin with liver health as well as with glucose and lipid metabolism markers in subjects with NAFLD. Current findings led to the identification of ferritin as a potential noninvasive predictive biomarker of NAFLD, whose surrogate value increased when combined with other routine biochemical measurements (glucose/ALT).

## 1. Introduction

Nonalcoholic fatty liver disease (NAFLD) is a condition defined by an excessive triglyceride accumulation in liver cells that is not caused by heavy alcohol consumption [1]. NAFLD is a worldwide major cause of liver disease [2] which potentially contributes to a burden of extrahepatic disturbances. Indeed, NAFLD is considered a multiorgan failure linked to obesity, cardiovascular disease (CVD), insulin resistance (IR), or metabolic syndrome (MetS) features [2,3,4]. This morbid condition can lead to nonalcoholic steatohepatitis (NASH), advanced fibrosis, cirrhosis, and finally, hepatocellular carcinoma [5]. Multiple environmental and genetic factors are involved in the onset and progression of NAFLD [6]. Concerning NAFLD treatments, weight loss induced by energy-restricted diets, physical activity promotion, and other lifestyle modifications have exhibited promising results leading to a better hepato-metabolic status [7,8]. Liver biopsy, the current reference standard, is an invasive and expensive procedure with some inherent surgical risks and only represents around 1/50,000 of the total hepatic volume [2,9]; however, it is still required for a definite diagnosis of NASH. In this context, noninvasive liver biomarkers and reproducible surrogate routine laboratory tests are sought as feasible alternatives to liver biopsy. Therefore, research is focusing on more efficient diagnostic and predictive biomarkers for identifying NAFLD features at early stages [2,9,10,11].

Novel investigations evidenced that iron metabolism-related parameters may be suitable predictors of liver disease outcomes [12]. The liver is the major iron storage organ and plays a key role in the metabolism of this nutrient [13]. Thus, iron has been involved in cellular oxidative stress and IR, key features of NAFLD pathogenesis, and hepatic iron accumulation has been linked to advanced fibrosis [14,15]. Ferritin is the chief iron storage protein but also is an acute-phase protein, and serum concentrations are increased in inflammatory conditions [16]. In this context, it remains unclear if serum ferritin reflects liver damage and accompanying inflammation features, increased body iron stores, or a combination of these factors [17]. Mild-to-moderate serum ferritin levels have been related to higher risk of NAFLD [18,19], and increased ferritin levels have been associated to more advanced NAFLD and higher mortality risk [20].

In this context, the objective of this research was to explore the usefulness of ferritin as a predictive surrogate biomarker of NAFLD condition, alone or in combination with other routine biochemical parameters.

## 2. Materials and Methods

### 2.1. Participants

The current research is an ancillary cross-sectional analysis including baseline data of the FLiO study (Fatty Liver in Obesity), a randomized controlled trial (www.clinicaltrials.gov; NCT03183193). The study included 112 (65 Male and 47 Female) adults (aged 40–80 years) with overweight/obesity (body mass index (BMI) ≥27.5 kg/m^2^ to <40 kg/m^2^). The presence of hepatic steatosis was determined by ultrasonography, and then hepatic fat was quantified by magnetic resonance imaging (MRI). The biopsy procedure was not performed.

Exclusion criteria included the presence of known liver disease (other than NAFLD), ≥3 kg body weight loss in the last 3 months, elevated alcohol consumption (>21 and >14 units of alcohol per week for men and women, respectively) [21], endocrine disorders (hyperthyroidism or uncontrolled hypothyroidism), pharmacological treatment with immunosuppressants, cytotoxic agents, systemic corticosteroids (or other drugs that could potentially cause hepatic steatosis or alteration of liver tests), active autoimmune diseases or requiring pharmacological treatment, acute infections, the use of weight modifiers, the presence of severe psychiatric disorders, and inability to follow the diet (food allergies, intolerances) as well as difficulties to follow scheduled visits.

All the procedures performed have been complied with all the relevant national regulations, institutional policies, and in accordance the tenets of the Helsinki Declaration. The study protocol and informed consent document were approved by the Research Ethics Committee of the University of Navarra on 24 April, 2015 (ref. 54/2015). All individuals gave written informed consent prior to inclusion.

### 2.2. Anthropometrics, Body Composition, and Biochemical Assessment

Anthropometric measurements (body weight, height, and waist circumference), body composition (DXA, Lunar iDXA, encore 14.5, Madison, WI, USA), and blood pressure (Intelli Sense. M6, OMRON Healthcare, Hoofddorp, The Netherlands) were determined in fasting state following standardized procedures [22]. BMI was calculated as weight (kg) divided by the square of height (m).

Blood samples were properly collected at baseline after 12 h overnight fast, processed (15 min; 3500 rpm; 5 °C), and stored at −80 °C until the biochemical analyses were performed. Blood glucose, glycated hemoglobin (HbA1c), total cholesterol (TC), high-density lipoprotein cholesterol (HDL-c), triglycerides (TG), alanine aminotransferase (ALT), aspartate aminotransferase (AST), and gamma glutamyl transferase (GGT) were determined on an autoanalyzer with specific commercial kits and following the instructions of the company (cobas 8000, Roche Diagnostics, Basel, Switzerland). Insulin, adiponectin, fibroblast growth factor 21 (FGF-21), retinol-binding protein 4 (RBP-4), and dipeptidyl-peptidase 4 (DPP4) concentrations were quantified using specific ELISA kits (Demeditec; Kiel-Wellsee, Kiel, Germany) in a Triturus autoanalyzer (Grifols, Barcelona, Spain). Serum ferritin levels were analyzed by an external certified laboratory (Eurofins Megalab S.A, Madrid, Spain) using a chemiluminescent microparticle immunoassay (CMIA) technology (Abbott Architect Ferritin Assay). The low-density lipoprotein (LDL-c) levels were calculated using the Friedewald formula [23]: LDL-c = TC − HDL-c − TG/5. The Homeostatic Model Assessment of Insulin Resistance (HOMA-IR) [24], the triglyceride-glucose (TyG) index (LnTG (mg/dL) x glucose (mg/dL)/2) [25], and the TG/HDL-c index (TG (mg/dL)/HDL-c (mg/dL)) were also calculated as described elsewhere [26].

Physical activity was classified in four different categories (sedentary, mild, moderated, or elevated).

### 2.3. Hepatic Imaging Techniques

The entire hepatic assessment was determined under fasting conditions by qualified staff at the University of Navarra Clinic. The presence of hepatic steatosis was determined by ultrasonography (Siemens ACUSON S2000 and S3000) in accordance with the previously described methodology [27]. Magnetic resonance imaging (MRI) (Siemens Aera 1.5 T) was performed to quantify the fat and iron content of the liver and the hepatic volume (HISTO technique), as described elsewhere [28]. 

### 2.4. Dietary Intake Estimate

Dietary intake was assessed with a validated semiquantitative 137-item food frequency questionnaire (FFQ) as described elsewhere [29]. Each item in the questionnaire included a typical portion size. For each food item, daily food consumption was estimated by multiplying the portion size by the consumption frequency and dividing as described elsewhere [30]. The nutrient composition of the food items was derived from accepted Spanish food composition tables.

The dietary total antioxidant capacity (TAC) score was calculated by computing the individual TAC values from the ferric reducing antioxidant power assay of each food. The mean TAC value of the foods contained in each item was used to calculate the dietary TAC score from the FFQ [31].

The adherence to the Mediterranean diet was assessed with a 17-point screening questionnaire, with a final score ranging from 0 to 17 and a higher score indicating a better adherence to the Mediterranean diet [32].

Glycemic index (GI) values for single food items on the food frequency questionnaire were derived from the “International Tables of Glycemic Index and Glycemic Load Values” as previously reported [31]. Total dietary GI was estimated by multiplying the amount of available carbohydrate (g) of each food item by its GI. The sum of these products was divided by the total carbohydrate intake. Glycemic load (GL) was also calculated, which represents the amount of carbohydrates multiplied by the average GI [31].

### 2.5. Statistical Analyses

The normal distribution of the continuous variables was assessed using the Shapiro-Wilk test. The data were expressed as a mean ± standard deviation for continuous traits and percentage for categorical variables. Participants were classified according to sex-specific serum ferritin tertiles (women: T1: <31.8, T2: ≥33.5 to <76.6, and T3: ≥80.4; men: T1: <109.8, T2: ≥116.1 to <263.7, and T3: ≥272.1). Differences in anthropometric data, body composition, biochemical variables, hepatic status, and dietary characteristics among the three ferritin sex-specific tertiles were tested by the nonparametric counterpart test (Kruskal-Wallis) and the χ2 test for categorical variables. Spearman correlations were performed to further explore the association between serum ferritin levels and both liver and glucose state. Multivariable adjusted quantile regression models were performed to evaluate the association between serum ferritin (in tertiles) and liver status variables (ALT, liver fat mass, and hepatic iron content). We ran first a minimally adjusted Model 1 (age and sex). Model 2 was adjusted by age, sex, Mediterranean diet adherence score, physical activity, and BMI. Model 3 was adjusted for age, sex, meat consumption, physical activity, and BMI. Model 4 was adjusted for age, sex, meat consumption, physical activity and HOMA-IR. Multivariable adjusted Model 5 was adjusted for age, sex, meat consumption, physical activity and DPP4 and finally Model 6 was adjusted for age, sex, meat consumption, physical activity and RBP4. Receiver operating characteristic (ROC) curves were applied to calculate the power of prediction of serum ferritin for liver fat and hepatic iron content (NAFLD). Also, combination panels were created to calculate the power of prediction including glucose, ALT, and TG. Validation of these results was performed calculating the optimism-corrected value using the Tibshirani’s enhanced bootstrap method described by Harrell [33].

Statistical analyses were performed using Stata version 12.1 (StataCorp 2011, College Station, TX, USA). All *p*-values presented are two-tailed, and differences were considered statistically significant at *p* < 0.05.

## 3. Results

The average age of study subjects was 51 ± 9 years old, and 42% were women. The mean BMI of the participants was 34 ± 4 kg/m^2^, with a waist circumference of 110 ± 8 cm. Subjects were categorized according to serum ferritin sex-specific tertiles. An overview on anthropometric data, body composition, glucose and lipid metabolism, liver markers, and dietary characteristics, considering serum ferritin tertiles, is given in Table 1 and Table 2, respectively.

Anthropometric and body composition variables showed no mentionable statistical differences among serum ferritin groups. No significant differences were observed in any glucose or lipid marker among ferritin tertiles. Regarding liver health status, participants in the third ferritin tertile had increased ALT, AST, and GGT concentrations and higher liver fat mass and hepatic iron content than subjects from the other groups (*p* < 0.05) (Table 1).

Concerning dietary characteristics, no statistically significant differences were observed in total energy intake and macronutrient distribution among serum ferritin tertiles (Table 2). When food groups were evaluated, main differences were observed in meat, whose consumption was increased in participants with higher serum ferritin (*p* < 0.05). On the other hand, subjects from the third tertile consumed less fish than subjects from the other two tertiles (*p* < 0.05) (Table 2). No differences were shown in dietary quality indicators (GI, GL, and TAC), although a tendency in the Mediterranean dietary score was observed among serum ferritin tertiles since the adherence to the Mediterranean diet pattern reduced as serum ferritin increased (Table 2).

Further analyses were performed regarding dietary intake and food group consumption. In addition to the previous results, fruit consumption and adherence to the Mediterranean diet were inversely proportional to the levels of serum ferritin.

A subanalysis concerning sex was performed in order to evaluate the effect of sex in the link between serum ferritin and variables of interest (Table A1 and Table A2). Remarkably, stronger associations of ferritin levels with glucose, lipid, and liver status were found in men. Men above the serum ferritin median had significantly higher triglyceride levels, TyG, and TG/HDL-c indices, as well as lower HDL-c concentration, than men below the median. Men above the serum ferritin median also registered significantly higher transaminase levels, liver fat, and iron content compared with men below the median (Table A1). Concerning dietary features, statistically significant differences were observed only among men above and below the ferritin median. Men above the serum ferritin median consumed more meat and less fruits and fish than men below the ferritin median. A higher adherence to the Mediterranean diet pattern was observed in those men whose ferritin levels were below the median. This sample also registered lower dietary GI values (Table A2). In women, the significant differences disappeared although the same trends were maintained when analyzing metabolic and nutritional status (Table A1 and Table A2).

The link between serum ferritin levels and liver, lipid, and glucose metabolism was further explored. Positive associations of serum ferritin with HOMA-IR and TyG index were found concerning glucose metabolism (Figure 1 and Table A3). When lipid parameters were evaluated, positive correlations of serum ferritin concentrations with TG and TG/HDL index were observed whereas HDL-c was negatively associated with ferritin. Regarding hepatic status, serum ferritin was positively correlated with ALT, AST, GGT, hepatic fat, liver iron, hepatic volume, and steatosis degree (Figure 1 and Table A3). When analyzing cytokines, significant positive associations of ferritin with DPP4 and RBP-4 were observed (Figure 1 and Table A3).

Multivariable quantile regression models were performed with NAFLD markers (ALT, liver fat mass, and liver iron) as dependent factors and serum ferritin (in tertiles) as the independent variable (Table 3). Minimally adjusted (Model 1: age and sex) and multiple adjusted (Model 2: age, sex, Mediterranean diet adherence score, physical activity, and BMI; Model 3: age, sex, meat consumption, physical activity, and BMI; Model 4: age, sex, meat consumption, physical activity, and HOMA-IR; Model 5: age, sex, meat consumption, physical activity, and DPP4; Model 6: age, sex, meat consumption, physical activity, and RBP4) models exhibited positive associations between the lowest to highest tertile of serum ferritin concentrations and ALT, liver fat mass, and hepatic iron content.

In order to further analyze the potential usefulness of serum ferritin as a predictor of NAFLD, the receiver operating characteristic (ROC) curves for ferritin were calculated, using the MRI technique as the reference method to quantify the liver fat and hepatic iron. The areas under the curve (AUC) of serum ferritin were 0.73 and 0.68 for liver fat and hepatic iron content, respectively. We also investigated whether its combination with other biochemical parameters might improve the AUC of serum ferritin alone. Forward-selection procedures identified the combination of ferritin, glucose, and ALT (AUC 0.82) as the best predictive score for liver fat mass, followed by a combination panel formed of ferritin and glucose (AUC 0.80). On the other hand, a panel combination of ferritin and ALT showed the major predictive ability for liver iron content (AUC 0.73), followed by a panel designed with ferritin and TG (AUC 0.72) (Figure 2). Validation of these results was performed by calculating the optimism-corrected value using the Tibshirani’s enhanced bootstrap method described by Harrell [33]. Results showed valuable AUCs (Figure 2).

## 4. Discussion

The current research involving the Fatty Liver in Obesity (FLiO) project shows the association of serum ferritin concentration with liver health as well as glucose and lipid metabolism in participants with NAFLD. The analysis of ferritin by means of quantile regression showed a positive association with ALT, liver fat content, and hepatic iron. Our data have also driven to assess ferritin as a predictive biomarker of NAFLD. Remarkably, serum ferritin allowed predicting the liver fat deposition and hepatic iron content by MRI, alone or in combination with other routine biochemical parameters such as TG, ALT, and glucose.

NAFLD is a clinical syndrome increasing globally, and it is a leading cause of chronic liver disease [2]. The liver is the major site of systemic iron regulation [34]. Hepatocytes constitute the major parenchymal iron storage pool and contain large amounts of ferritin, the primary iron storage protein [35]. Iron is an essential but potentially toxic element that may promote the onset and progression of NAFLD by increasing oxidative stress and altering insulin signaling and lipid metabolism [14,15,36,37]. Iron overload is observed in approximately one-third of adults with NAFLD [38]. In the present study, those participants from the third tertile according to ferritin levels showed higher liver iron storage as well as higher liver fat accumulation and increased transaminases concentrations. Moreover, ferritin levels were strongly related to liver iron percentage. In line with our results, Ryan et al. reported a strong association between ferritin and hepatic iron content by MRI in 129 participants with NAFLD [39]. Scientific evidences have shown that increased iron stores are intimately connected to β-cell dysfunction, impaired glucose metabolism, type 2 diabetes, DNA damage, and lipid peroxidation [24,36,37]. The main mechanism proposed is that iron promotes oxidative stress reactions resulting in cellular damage [14,15]. Indeed, recent data suggest that iron-induced reactive oxygen species (ROS) initiate an oxidative stress cascade causing lipid peroxidation and disturbances in insulin signaling. Increased free radicals might contribute to insulin resistance via increased free fatty acids oxidation, reduction of glucose uptake by the muscle, and impaired insulin release [40,41]. At the same time, the damage produced to hepatic cells might induce an increase in circulating ferritin concentration [16]. In addition to this, ferritin is an acute-phase reactant, and the low-grade inflammatory state induced by obesity as well as NAFLD might also cause the increase in serum ferritin concentration [17,18]. Serum ferritin was strongly associated with ALT and liver fat content, suggesting a close connection between high serum ferritin levels and impaired liver metabolism.

Interestingly, we found that serum ferritin was positively associated with HOMA-IR, a marker of IR. As a novelty, serum ferritin levels were positively related with DPP4 and RBP4, giving new molecular pathways that could explain the link between iron homeostasis and IR. Scientific evidences have shown that ferritin is associated with reduced adiponectin concentration, a key mediator of insulin sensitivity [42]. In this sense, we did not find an association between ferritin and adiponectin. On the other hand, the association between ferritin and liver markers remained significant after adjustment for IR (HOMA-IR, RBP4, and DPP4), suggesting that the relationship between ferritin and liver status is not entirely explained by alterations induced in glucose and insulin metabolism, but also other metabolic pathways seem to be involved.

About lipid metabolism, serum ferritin was significantly related to high triglycerides and low HDL-c levels. In line with our results, there is a growing body of evidence that iron may affect lipid metabolism, possibly via hepcidin [43]. Some researchers reported a positive association between hepatic hepcidin expression and TC, TG, and LDL-c concentrations in NAFLD [44]. In a meta-analysis, Suárez-Ortegón et al. evaluated the association between ferritin and MetS. Remarkably, they reported that high triglycerides and glucose were the components more strongly linked to ferritin [45]. Additionally, numerous proteomic and hepatic gene expression studies have found a link between iron homeostasis and lipid status, although more research is needed to further elucidate this relationship in the context of NAFLD and progression to NASH [43,44,46].

When dietary intake and food groups were explored, we evidenced that meat consumption was increased in participants with higher serum ferritin. These results were in accordance with the literature, since numerous evidences have suggested that some meat components such as heme-iron, sodium, and preservatives could be potentially harmful for health and, specifically, liver function [47,48]. In this context, some studies found an association of meat or heme-iron intake with higher serum ferritin, leading to necroinflammation and fibrosis, both hallmarks of NAFLD [49]. On the other hand, fish was associated with lower concentrations of ferritin. In this context, the omega-3 polyunsaturated fatty acids (PUFAs) contained in fatty fish might exert beneficial effects over ferritin levels. Research studies have shown that omega-3 PUFAs are inversely associated with NAFLD, by decreasing proinflammatory molecules, TG, and improving liver histology [43,50]. In addition, fish contains lower heme-iron when compared with red meat, which might explain the results obtained in this study [51]. Fish could be proposed as a healthier dietary alternative whereas meat consumption should be controlled in the management of NAFLD.

Currently research is focused on more efficient diagnostic and predictive biomarkers for identifying NAFLD features at early stages [2,10,11,12,52], trying to replace liver biopsy. Recent studies evidenced that iron metabolism-related parameters may be suitable predictors of liver disease outcomes [13]. In this research, we hypothesized that serum ferritin might constitute a marker of fatty liver in subjects with NAFLD. Serum ferritin concentration seems to be a good biomarker intimately connected to liver health condition, allowing the prediction of hepatic fat and iron content, alone or in combination with other routine biochemical parameters. Indeed, the combination of ferritin, glucose, and ALT showed the best prediction for liver fat mass with an accuracy of 80%. On the other hand, a panel combination of ferritin and ALT showed the major predictive ability for liver iron content (AUC 0.73). The internal validation of these ROC analyses strengthens the obtained result; however, more studies should be performed to identify and validate robust noninvasive tests to help in the identification of subjects with NAFLD and subjects at risk for the development of the disease [7,11].

This assay adds further insights and knowledge about the link between iron metabolism and NAFLD. Serum ferritin levels showed a relevant impact on both liver health and general metabolism, being a key factor to be considered in the management of NAFLD. Our results also suggest the possible clinical use of ferritin as an indicator of NAFLD, alone or in combination with other routine biochemical measures. The design of different predictive models for NAFLD through blood biomarkers has many advantages, although further investigation and consensus are needed.

The current study presents some limitations. Firstly, the cross-sectional nature of the study does not allow the establishment of causality. Thus, longitudinal studies are needed to determine whether ferritin might be a good predictor of the progression of the disease or if it is just a consequence of the liver function alteration. Secondly, the presence of hepatic steatosis was determined by ultrasonography. Hepatic fat was quantified by magnetic resonance imaging (MRI), and the biopsy procedure was not performed. Thirdly, dietary data were evaluated using self-reported information of the participants, and thus, the results are susceptible to some degree of bias. Fourthly, other iron metabolism parameters such as transferrin, serum iron, or hepcidin were not determined and could provide complementary information. On the other hand, some strength can be mentioned. Participants have been carefully selected following exclusion and inclusion criteria to avoid a heterogeneous sample. Liver disease was assessed by qualitative (ultrasonography) and quantitative (MRI) methodology in order to achieve a good liver health characterization. Dietary questionnaires were revised by a qualified dietician in order to diminish possible fill-in errors.

## 5. Conclusions

The present study demonstrated the association of serum ferritin with liver health (ALT, liver fat content, and hepatic iron) as well as glucose and lipid metabolism in individuals with NAFLD. Additionally, this research identified ferritin as a potential biomarker of NAFLD, enabling to predict the liver fat deposition and hepatic iron content by MRI, alone or in combination with other routine biochemical parameters such as TG, ALT, and glucose.

## Figures and Tables

**Figure 1 diagnostics-10-00917-f001:**
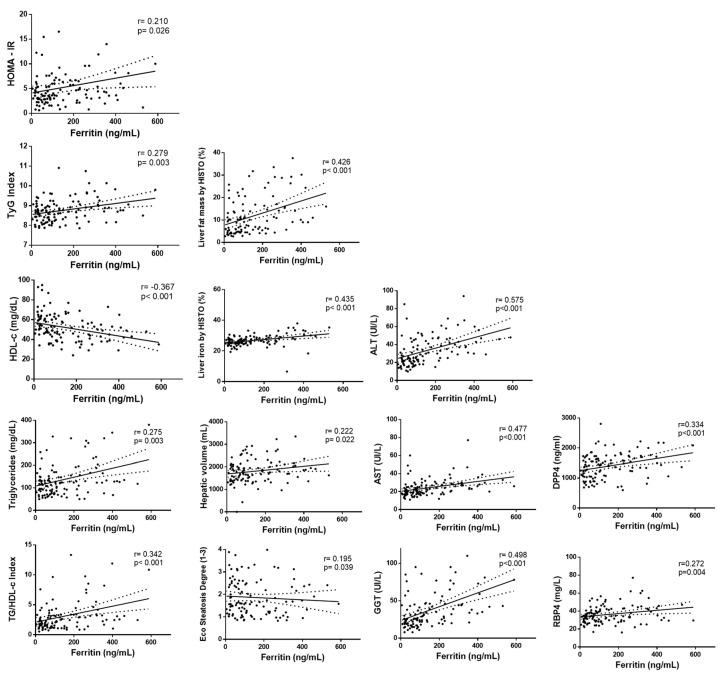
Correlation analyses between serum ferritin levels with liver status and glucose metabolism-related markers. First column: HOMA-IR, TyG index, HDL-c, triglycerides, TG/HDL-c. Second column: liver fat mass, liver iron, hepatic volume, steatosis degree. Third column: ALT, AST, GGT. Fourth column: DPP4 (dipeptidyl-peptidase 4), RBP-4 (retinol-binding protein 4).

**Figure 2 diagnostics-10-00917-f002:**
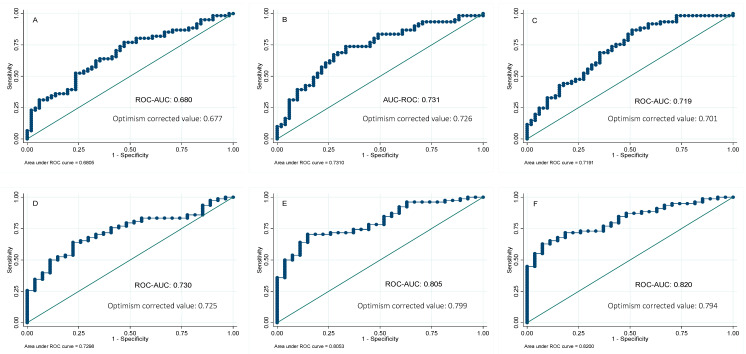
Receivers operating curves between liver iron percentage by magnetic resonance imaging (MRI) and (**A**) ferritin; (**B**) ferritin and ALT; (**C**) ferritin and triglycerides. Receivers operating curves between liver fat percentage by magnetic resonance imaging (MRI) and (**D**) ferritin; (**E**) ferritin and glucose; (**F**) ferritin, glucose, and ALT. AUC, area under the ROC curve. Reference line.

**Table 1 diagnostics-10-00917-t001:** Baseline characteristics of participants according to sex-specific serum ferritin tertiles.

Characteristics	Overall (*n* = 112)	Serum Ferritin Tertiles			*p*-Value
		T1 (*n* = 38)	T2 (*n* = 37)	T3 (*n* = 37)	
Serum ferritin level (ng/mL)	100.7 (51; 226)	5.3 to <68.7	68.7 to <177.1	177.1 to <588.1	
Age (years)	51.0 (45; 56)	53 (48; 64)	47 (43; 52) *	51 (46; 56)	0.006
Sex (male/female)	65/47	22/16	22/16	21/15	0.999
BMI (kg/m^2^)	33.5 (31; 36)	34.0 (32; 36)	33.6 (31; 36)	31.9 (31; 35)	0.152
Physical activity (%)					
Never	41.1	47.4	34.2	41.7	0.440
Mild	23.2	26.3	21.0	22.2	
Moderated	23.2	13.2	31.6	25.0	
Elevated	12.5	13.2	13.2	11.1	
Cardiometabolic Risk Factors					
Waist circumference (cm)	108.9 (104; 116)	114.1 (105; 118)	108.8 (101; 115)	106.8 (102; 112)	0.142
Total fat mass (kg)	38.2 (33; 44)	39.7 (35; 46)	39.6 (34; 45)	35.4 (32; 41)	0.128
Visceral fat mass (kg)	2.2 (2; 3)	2.3 (2; 3)	2.1 (2; 3)	2.1 (1; 3)	0.399
Systolic blood pressure (mmHg)	130.0 (122; 142)	132.0 (124; 143)	130.0 (121; 142)	127.5 (120; 139)	0.433
Diastolic blood pressure (mmHg)	87.0 (80; 92)	87.5 (82; 93)	88.0 (83; 92)	85.0 (79; 91)	0.493
Glucose Metabolism Variables					
Glucose (mg/dL)	102.0 (93; 111)	103.5 (97; 112)	97.5 (91; 111)	100.5 (91; 109)	0.217
Insulin (mU/L)	16.4 (11; 23)	16.5 (12; 21)	13.1 (9; 21)	19.3 (13; 25)	0.097
HbA1c (%)	5.6 (5; 6)	5.7 (5; 6)	5.5 (5; 6)	5.6 (5; 6)	0.081
HOMA-IR	4.1 (3; 6)	4.2 (3; 6)	3.3 (2; 6)	4.6 (3; 7)	0.102
TyG index	8.8 (8; 9)	8.7 (8; 9)	8.7 (8; 9)	8.7 (8; 9)	0.873
TG/HDL-c index	2.4 (1; 3)	2.2 (2; 3)	2.5 (1; 3)	2.5 (1; 3)	0.851
Lipid Metabolism Variables					
Total cholesterol (mg/dL)	199.5 (167; 225)	185.5 (159; 234)	210.0 (161; 232)	201.5 (178; 213)	0.847
LDL cholesterol (mg/dL)	118.4 (91; 143)	113.8 (93; 145)	120.1 (86; 144)	128.2 (99; 138)	0.948
HDL cholesterol (mg/dL)	50.0 (43; 61)	52.0 (46; 61)	47.5 (41; 64)	49.0 (40; 58)	0.417
Triglyceride (mg/dL)	123.0 (82; 156)	121.0 (91; 150)	123.0 (89; 147)	123.5 (76; 172)	0.992
Liver Status Variables					
ALT (IU/L)	29.0 (21; 43)	23.0 (19; 28)	29.5 (18; 46)	38.5 (29; 47) #†	<0.001
AST (IU/L)	22.5 (18; 28)	20.5 (17; 25)	20.5 (17; 28)	26.5 (21; 33) #†	0.003
GGT (IU/L)	29.0 (20; 45)	25.0 (19; 34)	29.5 (20; 46)	39.0 (22; 62) #	0.043
Liver fat mass (%)	9.4 (5; 16)	9.4 (5; 13)	6.0 (4; 11)	14.2 (9; 22) #†	0.002
Liver iron (%)	26.6 (25; 29)	26.0 (25; 27)	26.1 (25; 28)	28.6 (26; 32) #†	0.006
Hepatic volume (mL)	1791 (1425; 2078)	1922 (1397; 2131)	1676 (1425; 1956)	1701 (1474; 2054)	0.480
Eco steatosis degree (1–3 range)	1.6 (1; 2)	1.6 (1; 2)	1.7 (1; 3)	1.6 (1; 2)	0.493

Values are represented as median (interquartile range). Abbreviations: BMI: body mass index; HbA1c: glycosylated hemoglobin; HOMA-IR: homeostatic model assessment of insulin resistance; TyG index: triglyceride-glucose index; TG/HDL-c index: triglyceride/high-density lipoprotein cholesterol index; ALT: alanine aminotransferase; AST: aspartate aminotransferase; GGT: gamma-glutamyl transferase. * *p* was significant between T1 and T2; # *p* was significant between T1 and T3; † *p* was significant between T2 and T3.

**Table 2 diagnostics-10-00917-t002:** Description of the nutrient and food consumption according to sex-specific serum ferritin tertiles.

Characteristics	Overall (*n* = 112)	Serum Ferritin Tertiles			*p*-Value
		T1 (*n* = 38)	T2 (*n* = 37)	T3 (*n* = 37)	
Serum ferritin level (ng/mL)	100.7 (51; 226)	5.3 to <68.7	68.7 to <177.1	177.1 to <588.1	
Nutrients					
Energy intake (kcal/day)	2559 (2057; 3085)	2541 (2031; 3241)	2650 (1941; 3001)	2548 (2173; 3085)	0.866
Carbohydrates (%E)	43.3 (39; 48)	44.4 (39; 48)	42.0 (38; 45)	45.2 (40; 49)	0.118
Proteins (%E)	16.8 (15; 19)	15.8 (14; 19)	17.8 (15; 20)	16.7 (15; 19)	0.113
Lipids (%E)	36.5 (32; 42)	37.0 (31; 41)	37.7 (34; 43)	34.7 (33; 42)	0.506
Fiber (g/day)	24.1 (19; 30)	24.7 (19; 31)	24.3 (18; 29)	23.2 (18; 28)	0.586
Alcohol intake (g/day)	5.4 (1; 12)	6.8 (1; 12)	4.4 (1; 11)	4.8 (1; 13)	0.686
Iron (mg/day)	17.1 (15; 21)	16.9 (15; 22)	17.2 (14; 21)	17.6 (15; 21)	0.809
Vitamin C (mg/day)	180.1 (133; 242)	187.0 (134; 247)	181.0 (151; 264)	165.1 (101; 227)	0.276
Food Groups					
Vegetables (g/day)	283.6 (205; 380)	270.2 (220; 348)	303.1 (212; 402)	277.3 (161; 375)	0.602
Fruits (g/day)	246.3 (142; 454)	262.1 (165; 504)	290.7 (163; 468)	190.1 (127; 357)	0.206
Legumes (g/day)	16.8 (12; 25)	16.8 (12; 25)	16.5 (12; 25)	20.5 (16; 25)	0.812
Cereals (g/day)	197.9 (83; 228)	198.8 (75; 222)	197.7 (83; 221)	198.0 (101; 235)	0.602
Dairy products (g/day)	308.7 (234; 466)	308.7 (231; 435)	314.4 (254; 484)	314.8 (227; 530)	0.750
Meat (g/day)	187.9 (139; 237)	151.7 (111; 211)	210.2 (147; 256) *	206.2 (140; 237) #	0.012
Fish (g/day)	88.5 (60; 122)	104.3 (69; 136)	86.6 (60; 135)	76.9 (53; 101) #	0.033
Nuts (g/day)	6.0 (2; 15)	4.2 (0; 25)	6.0 (2; 15)	6.0 (2; 12)	0.999
Dietary Quality Indices					
Glycemic index	54.7 (49; 59)	54.3 (47; 59)	53.4 (50; 57)	56.3 (51; 59)	0.234
Glycemic load	153.2 (101; 194)	158.8 (91; 203)	140.2 (93; 185)	153.8 (125; 192)	0.543
Total antioxidant capacity (mmol/day)	9.7 (8; 13)	9.6 (8; 12)	10.5 (8; 14)	9.7 (8; 13)	0.773
Mediterranean diet score (points)	6.0 (5; 7)	6.5 (5; 8)	6.0 (5; 7)	6.0 (4; 7)	0.057

Values are represented as median (interquartile range). * *p* was significant between T1 and T2; # *p* was significant between T1 and T3; *p* was significant between T2 and T3.

**Table 3 diagnostics-10-00917-t003:** Quantile regression models with nonalcoholic fatty liver disease (NAFLD) markers as dependent factors and serum ferritin tertiles as the independent variable among study participants.

Variables	Model 1 β (95% CI)	*p*-Value	Model 2 β (95% CI)	*p*-Value	Model 3 β (95% CI)	*p*-Value	Model 4 β (95% CI)	*p*-Value	Model 5 β (95% CI)	*p*-Value	Model 6 β (95% CI)	*p*-Value
ALT level (*n* = 112)												
Serum ferritin level (ng/mL)												
T1 (5.3-68.7)	1.00 (ref.)		1.00 (ref.)		1.00 (ref.)		1.00 (ref.)		1.00 (ref.)		1.00 (ref.)	
T2 (68.7–177.1)	6.00 (−1.19; 13.19)	0.101	5.28 (−1.66; 12.22)	0.134	4.79 (−3.11; 12.69)	0.232	5.66 (−2.62; 13.96)	0.178	5.38 (−2.28; 13.04)	0.167	5.40 (−2.58; 13.38)	0.183
T3 (177.1–588.1)	18.76 (10.70; 26.82)	<0.001	22.10 (13.89; 30.31)	<0.001	20.76 (11.79; 29.74)	<0.001	20.57 (11.07; 30.07)	<0.001	19.21 (10.39; 28.03)	<0.001	21.19 (12.14; 30.24)	<0.001
Liver fat (*n* = 112)												
Serum ferritin level (ng/mL)												
T1 (5.3–68.7)	1.00 (ref.)		1.00 (ref.)		1.00 (ref.)		1.00 (ref.)		1.00 (ref.)		1.00 (ref.)	
T2 (68.7–177.1)	4.06 (−1.27; 9.39)	0.134	2.82 (−2.0; 7.66)	0.249	2.84 (−2.31; 8.00)	0.277	2.41 (−1.71; 6.54)	0.249	3.69 (−1.94; 9.32)	0.196	2.22 (−2.59; 7.04)	0.361
T3 (177.1–588.1)	9.09 (3.19; 14.99)	0.003	9.81 (4.16; 15.47)	0.001	10.04 (4.29; 15.78)	0.001	6.42 (1.77; 11.08)	0.007	8.70 (2.37; 15.03)	0.008	8.71 (3.35; 14.06)	0.002
Liver iron (*n* = 112)												
Serum ferritin level (ng/mL)												
T1 (5.3–68.7)	1.00 (ref.)		1.00 (ref.)		1.00 (ref.)		1.00 (ref.)		1.00 (ref.)		1.00 (ref.)	
T2 (68.7–177.1)	1.61 (−0.44; 3.66)	0.122	1.08 (−0.91; 3.06)	0.285	1.16 (−0.52; 2.85)	0.174	1.11 (−0.65; 2.88)	0.212	1.26 (−0.45; 2.97)	0.147	0.84 (−1.18; 2.86)	0.410
T3 (177.1–588.1)	3.44 (1.18; 5.71)	0.003	2.61 (0.29; 4.93)	0.028	3.69 (1.81; 5.57)	<0.001	3.72 (1.73; 5.71)	<0.001	3.76 (1.84; 5.69)	<0.001	4.20 (1.96; 6.45)	<0.001

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
