# Peer review of "Predictive Value of Serum Ferritin in Combination with Alanine Aminotransferase and Glucose Levels for Noninvasive Assessment of NAFLD: Fatty Liver in Obesity (FLiO) Study"

_diagnostics, 2020, doi:10.3390/diagnostics10110917_

Round 1

Reviewer 1 Report

Galarregui et al investigated the ability of serum ferritin in the prediction of liver disease, namely fatty acids and iron liver deposition, in overweight and obese subjects. The authors reported that ferritin is a good predictor of fatty liver and iron content in their cohort. The study is well conducted and written. Statistical analyses support the findings and conclusions. I have some minor suggestions for the authors:

-NAFLD prevalence differs according to gender, it would be of interest to investigate whether the findings are different between males and females.

-The discussion section is too long, I suggest to shorten it.

Author Response

Response to Reviewer 1 Comments

We would like to thank all the reviewers for their thoughtful comments and suggestions. We believe that in addressing these comments, this revised manuscript is considerably improved. A detailed point-by-point response to the reviewer’s comments can be found after each question. Changes in the manuscript have been highlighted in colour.

Review of Manu Ref: diagnostics-982392

Comments and Suggestions for Authors

I found Galarregui et al investigated the ability of serum ferritin in the prediction of liver disease, namely fatty acids and iron liver deposition, in overweight and obese subjects. The authors reported that ferritin is a good predictor of fatty liver and iron content in their cohort. The study is well conducted and written. Statistical analyses support the findings and conclusions. I have some minor suggestions for the authors:

Point 1: NAFLD prevalence differs according to gender, it would be of interest to investigate whether the findings are different between males and females.

Response 1: As proposed by the reviewer, we have performed supplementary analyses regarding sex (Table A1; Table A2). Results show substantially the same trends that those achieved with the overall population. Remarkably, stronger associations of serum ferritin with hepatic, lipid and glucose status were found in men. In women, the significant differences disappeared although the same trends were maintained when analyzing liver and glucose metabolism markers.

We have included a paragraph concerning this matter in the results of the present research: “A sub-analysis concerning sex was performed in order to evaluate the effect of sex in the link between serum ferritin and variables of interest (Table A1; Table A2). Remarkably, stronger associations of ferritin levels with glucose, lipid and liver status were found in men. Men above the serum ferritin median had significantly higher triglyceride levels, TyG and TG/HDL-c indices than men below the median. HDL-c concentration was significantly lower in men above the ferritin median. Men above the serum ferritin median also registered significantly higher transaminase levels, liver fat and iron content compared to men below the median. Concerning dietary features, statistically significant differences were observed only among men above and below the ferritin median. Men above the serum ferritin median consumed more meat and less fruits and fish than men below the ferritin median. A higher adherence to the MedDiet pattern was observed in those men whose ferritin levels were below the median. This sample also registered lower dietary GI values. In women, the significant differences disappeared although the same trends were maintained when analyzing metabolic and nutritional status.” Lines 206-218.

Point 2: The discussion section is too long, I suggest to shorten it.

Response 2: As suggested by the reviewer, we have shortened the discussion.

Reviewer 2 Report

In the present manuscript, the authors analyzed a cohort of 112 well-characterized patients with NAFLD. Several routine laboratory parameters were investigated as well as serum ferritin levels, which were considered a potential proxy biomarker for liver iron metabolism and consequently liver damage. The authors demonstrate that ferritin levels are useful to predict the liver fat mass as determined by MRI and ultrasound, alone or in combination with other routine parameters just as ALT or glucose levels. Overall, the manuscript is scientifically sound and transparent in its way of data presentation. The results are of relevance and the assessment of ferritin levels into routine diagnostics seems to be realistic.

However, I have one major concern regarding the presented ROC curves, because I cannot find any information about correction of the respective AUCs. It is likely that the estimated sensitivities and specificities are overfitted because only one data set was analyzed. The authors should consider performing a statistical approach for cross-validation and correction of the respective values (e.g. leave one out x-validation).

I suggest that the authors present some of their data as figures. To my opinion, including a correlation matrix illustrating the results that are currently presented as Table 3 would be beneficial.

The manuscript might be a bit too comprehensive considering the scope of presented results. The discussion might be shortened. Some aspects which are slightly redundant with the introduction might be removed.

Author Response

We would like to thank all the reviewers for their thoughtful comments and suggestions. We believe that in addressing these comments, this revised manuscript is considerably improved. A detailed point-by-point response to the reviewer’s comments can be found after each question. Changes in the manuscript have been highlighted in colour.

Review of Manu Ref: diagnostics-982392

Comments and Suggestions for Authors

In the present manuscript, the authors analyzed a cohort of 112 well-characterized patients with NAFLD. Several routine laboratory parameters were investigated as well as serum ferritin levels, which were considered a potential proxy biomarker for liver iron metabolism and consequently liver damage. The authors demonstrate that ferritin levels are useful to predict the liver fat mass as determined by MRI and ultrasound, alone or in combination with other routine parameters just as ALT or glucose levels. Overall, the manuscript is scientifically sound and transparent in its way of data presentation. The results are of relevance and the assessment of ferritin levels into routine diagnostics seems to be realistic.

Point 1: However, I have one major concern regarding the presented ROC curves, because I cannot find any information about correction of the respective AUCs. It is likely that the estimated sensitivities and specificities are overfitted because only one data set was analyzed. The authors should consider performing a statistical approach for cross-validation and correction of the respective values (e.g. leave one out x-validation).

Response 1: Thank you for your comment. The presented ROC curves have been corrected and internal validated, calculating the optimism corrected value using the Tibshirani’s enhanced bootstrap method described by Harrell (Figure 2), which is now stated in the Methods section (statistics).

Point 2: I suggest that the authors present some of their data as figures. To my opinion, including a correlation matrix illustrating the results that are currently presented as Table 3 would be beneficial.

Response 2: Considering the author´s comment, we have now included most important correlations as figures (Figure 1), and the whole correlation matrix has been included as supplementary information (Table A3).

Point 3: The manuscript might be a bit too comprehensive considering the scope of presented results. The discussion might be shortened. Some aspects which are slightly redundant with the introduction might be removed.

Response 3: As suggested by the reviewer, we have shortened the discussion and removed some redundant information that appeared in the introduction.

Reviewer 3 Report

In the present study, Galarregui and colleagues investigated the possible role of ferritin levels as a biomarker of NAFLD, compared or combined with other biochemical routine parameters. The authors observed that ferritin levels were associated to necroinflammation (ALT values), hepatic iron and liver fat content. Interstingly, the combination of ferritin + glucose + ALT achieved a good diagnostic accuracy for the prediction of liver fat mass, while  ferritin + ALT showed a moderate performance for the estimation of liver iron content. Overall, the manuscript is well written and data are clearly presented. 

Below, some minor comments:

1) Introduction. Line 66-67. While imaging methods are sufficent to determine the presence of liver steatosis, in the setting of NAFLD, liver biopsy (showing steatosis, hepatocyte ballooning and lobular inflammation) is still required for a definite diagnosis of NASH. Authors should mention this aspect in the introduction section.

2) Methods. It is unclear to me whether patients underwent liver biopsy within the FLiO study.

3) Statistical analysis. Data normality was checked by Shapiro–Wilk test. Since all continuous variables have been reported as mean and SD, I assume that were all normally distributed (despite variables such as ALT and other biochemical parameters, rarely are normally distributed!). Therefore, all subsequent analysis should be performed using parametric test. Consistently, correlation analysis should be performed by Pearson rather than spearman correlation.

3) Results. Table 3. Authors may consider to add the estimates for r (i.e. the 95% CI)

4) Discussion. As mentioned above, liver biopsy is almost mandatory to precisely define NAFLD/NASH. Further, liver biopsy is the reference method for steatosis assessment (despite the important limitations of the procedure). Authors must add in the limitations paragraph of the discussion the lack of liver biospy (actually, if the patients have not been biosped).

5) Please check references. References from 51 onwards are not shown.

Author Response

We would like to thank all the reviewers for their thoughtful comments and suggestions. We believe that in addressing these comments, this revised manuscript is considerably improved. A detailed point-by-point response to the reviewer’s comments can be found after each question. Changes in the manuscript have been highlighted in colour.

Review of Manu Ref: diagnostics-982392

Comments and Suggestions for Authors

In the present study, Galarregui and colleagues investigated the possible role of ferritin levels as a biomarker of NAFLD, compared or combined with other biochemical routine parameters. The authors observed that ferritin levels were associated to necroinflammation (ALT values), hepatic iron and liver fat content. Interestingly, the combination of ferritin + glucose + ALT achieved a good diagnostic accuracy for the prediction of liver fat mass, while ferritin + ALT showed a moderate performance for the estimation of liver iron content. Overall, the manuscript is well written and data are clearly presented.

Below, some minor comments:

Point 1: Introduction. Line 66-67. While imaging methods are sufficient to determine the presence of liver steatosis, in the setting of NAFLD, liver biopsy (showing steatosis, hepatocyte ballooning and lobular inflammation) is still required for a definite diagnosis of NASH. Authors should mention this aspect in the introduction section.

Response 1: Thank you very much for your comment. As the reviewer suggests, we have mentioned this comment in the introduction: “Liver biopsy, the current reference standard, is an invasive and expensive procedure with some inherent surgical risks and only represents around 1/50000 of the total hepatic volume [2,9]; however it is still required for a definite diagnosis of NASH”. Lines 66-69.

Point 2: Methods. It is unclear to me whether patients underwent liver biopsy within the FLiO study.

Response 2: We have clarified that liver biopsy was not performed to the study participants: “The presence of hepatic steatosis was determined by ultrasonography and then hepatic fat was quantified by magnetic resonance imaging (MRI). The biopsy procedure was not performed.”. Lines 91-93.

Point 3: Statistical analysis. Data normality was checked by Shapiro–Wilk test. Since all continuous variables have been reported as mean and SD, I assume that were all normally distributed (despite variables such as ALT and other biochemical parameters, rarely are normally distributed!). Therefore, all subsequent analysis should be performed using parametric test. Consistently, correlation analysis should be performed by Pearson rather than Spearman correlation.

Response 3: Thank you for the comment. Variables in Tables 1/2 show baseline characteristics of the study participants. We have modified Table 1/2 and included the median and the interquartile range of all variables, as serum ferritin is a non-parametric variable. Non-parametric variables were analyzed using the non-parametric counterpart test (Kruskal-wallis). On the other hand, Spearman correlations are correctly applied instead of the Pearson analysis.

Point 4: Results. Table 3. Authors may consider adding the estimates for r (i.e. the 95% CI).

Response 4: As proposed by the reviewer, we have added the estimates for r (95% CI) when appropriate (Table A3).

Point 5: Discussion. As mentioned above, liver biopsy is almost mandatory to precisely define NAFLD/NASH. Further, liver biopsy is the reference method for steatosis assessment (despite the important limitations of the procedure). Authors must add in the limitations paragraph of the discussion the lack of liver biospy (actually, if the patients have not been biosped).

Response 5: Thank you for the comment. As the reviewer suggests, the lack of liver biopsy is a weakness of the study. This comment has been added to the limitations section: “Secondly, the presence of hepatic steatosis was determined by ultrasonography and then hepatic fat was quantified by magnetic resonance imaging (MRI). The biopsy procedure was not performed”.

Point 6: Please check references. References from 51 onwards are not shown.

Response 6: References have been completed.

Round 2

Reviewer 2 Report

Thank you for considering my suggestions.